

# Genome-wide characterization and expression analysis of soybean trihelix gene family

Wei Liu[1,2], Yanwei Zhang[1,2], Wei Li[1,2], Yanhui Lin[3], Caijie Wang[1,2], Ran Xu[1,2] and Lifeng Zhang[1,2]

[1] Crop Research Institute, Shandong Academy of Agricultural Sciences, Jinan, Shandong, China
[2] Shandong Engineering Laboratory of Featured Crops, Jinan, China
[3] Institute of Food Crops, Hainan Academy of Agricultural Sciences, Haikou, China

## ABSTRACT

Trihelix transcription factors play multiple roles in plant growth, development and various stress responses. In this study, we identified 71 trihelix family genes in the soybean genome. These trihelix genes were located at 19 out of 20 soybean chromosomes unevenly and were classified into six distinct subfamilies: GT-1, GT-2, GT$\gamma$, SIP1, SH4 and GT$\delta$. The gene structure and conserved functional domain of these trihelix genes were similar in the same subfamily but diverged between different subfamilies. Thirteen segmental duplicated gene pairs were identified and all of them experienced a strong purifying selective pressure during evolution. Various stress-responsive *cis*-elements presented in the promoters of soybean trihelix genes, suggesting that the trihelix genes might respond to the environmental stresses in soybean. The expression analysis suggests that trihelix genes are involved in diverse functions during soybean development, flood or salinity tolerance, and plant immunity. Our results provide genomic information of the soybean trihelix genes and a basis for further characterizing their roles in response to environmental stresses.

## INTRODUCTION

Transcription factors (TFs) are a type of DNA binding protein. They are capable of interacting with the *cis*-element of the promoter regions of the target genes, and regulate their expression. TFs play important roles in multiple development process and stress response in plants. Of all the reported TFs, the trihelix family is one of the first TFs discovered in plants (*Green, Kay & Chua, 1987*).

The trihelix family genes share one or two trihelix (helix-loop-helix-loop-helix) structures, each consisting of three putative-helices, which are responsible for binding to the GT motif (a light-responsive DNA element) (*Zhou, 1999*). Initial study of this family focused on the regulation of light-responsive genes (*Green, Kay & Chua, 1987*; *Zhou, 1999*). Subsequently, a majority of trihelix genes belonging to the different subfamilies have been cloned and characterized. Those trihelix genes were found to be involved in multiple development and growth processes such as: flowering, stomatal development,

Corresponding author
Lifeng Zhang,
zhanglifeng9639@sina.com

embryogenesis, and seed development (*Tzafrir et al., 2004*; *Breuer et al., 2009*; *Gao et al., 2009*; *Barr, Willmann & Jenik, 2012*; *Qin et al., 2014*). With the in-depth study of this gene family, more and more trihelix genes were proven to play important roles in abiotic and biotic stress resistance. The 3 rice (*Oryza sativa* L.) trihelix genes (*OsGTγ-1*, *OsGTγ-2* and *OsGTγ-3*) and two soybean (*Glycine max* (L.) Merr.) trihelix genes (*GmGT-2A* and *GmGT-2B*) were reported to be related to cold, drought, and salt stress response (*Xie et al., 2009*; *Fang et al., 2010*). The expression of Arabidopsis (A*rabidopsis thaliana* ) trihelix gene, *AtGT2L*, could be induced by cold and salt stresses (*Xi et al., 2012*); *GT-2 LIKE 1* (*GTL1*) loss-of-function mutations result in increased water deficit tolerance in Arabidopsis (*Yoo et al., 2010*). An oilseed rape (*Brassica napus*) gene *BnSIP1-1* played roles in ABA synthesis and signaling, as well as salt and osmotic stress response (*Luo et al., 2017*). In addition, it was also reported that the trihelix genes respond to waterlogging stress in maize (*Zea mays* L.) (*Du, Huang & Liu, 2016*). Meanwhile, the Arabidopsis gene *ASR3* could regulate the expression of genes related to immunity (*Li et al., 2015*).

As an important leguminous crop in the world, soybean provides an essential source of protein to the human diet, feed for live-stock, and as feedstock for the bio-diesel industry (*Wilcox, 2004*; *Koberg, Abu-Much & Gedanken, 2011*). However, soybean production can be dramatically decreased by the occurrence of environmental stresses, such as flooding (waterlogging or submergence) (*Ahmed et al., 2012*), high-salt (*Wang & Shannon, 1999*), as well as pathogen infection (*Tyler, 2007*). Several reports had performed the transcriptome analysis of soybean under abiotic and biotic stress (*Nanjo et al., 2011*; *Belamkar et al., 2014*; *Chen et al., 2016*; *Yin et al., 2016*). With the completion of soybean genome sequencing, many TF families including the abiotic/biotic stress-responsive transcription factor families have been genome-wide analyzed (*Song et al., 2018*; *Yu et al., 2016*; *Wang et al., 2019*). Considering the potential roles of trihelix family genes in stress tolerance, genome-wide analysis of soybean trihelix genes is necessary to be studied in depth.

In this study, we identified the soybean trihelix genes by the Myb/SANT-LIKE domain using HMM (Hidden Markow Model)-searches against a new version of the reference soybean genome (Wm82.a2. v1). We next analyzed their chromosomal distributions, gene duplication, motif composition, gene structure, *cis*-elements, phosphorylation sites, and miRNA targeting. Subsequently, we performed public-RNA-Seq data analysis to explore their tissue expression patterns. Furthermore, by analyzing the expression profile of trihelix genes in soybean suffering from submergence, high-salt and MAMP (microbe-associated molecular patterns) mixture treatment at the seedling stage, we identified several candidate trihelix genes that might contribute to the stress tolerance in soybean. Our results provide information of soybean trihelix genes and theoretical basis for their functional analysis, especially in the environmental stress responses.

## MATERIALS & METHODS

### Genome-wide identification of trihelix TFs in soybean

The genome sequences of soybean were downloaded from the Phytozome database (*Goodstein et al., 2011*) (v.12.1) (https://phytozome.jgi.doe.gov/pz/portal.html). The

Hidden Markow Model (HMM) of Myb/SANT-LIKE domain (PF13837) was retrieved from the Pfam database (http://pfam.xfam.org) (*Finn et al., 2015*). To identify candidate trihelix TFs, HMMER program was used to search the soybean genome with an *E*-value cutoff $<e^{-5}$. All putative proteins were confirmed by the Pfam and NCBI-CDD database (https://www.ncbi.nlm.nih.gov/cdd). The molecular weights and isoelectric points of soybean trihelix proteins were estimated using the online software ExPASy (https://www.expasy.org) (*Artimo et al., 2012*). Subcellular localization was predicted by using CELLO (V.2.5, http://cello.life.nctu.edu.tw/) (*Yu et al., 2006*).

## Multiple alignment and phylogenetic analysis

Multiple sequence alignments were performed using Clustalw with the amino acid sequences of trihelix family proteins. A neighbor-joining (NJ) phylogenetic tree was constructed using MEGA (v.7.0) (https://www.megasoftware.net) (*Kumar, Stecher & Tamura, 2016*) by the method with the following parameters: Poisson correction, pairwise deletion, and 1,000 bootstrap replicates.

## Gene structure analysis and identification of conserved motifs

The gene structures of trihelix genes were determined by the Gene Structure Display Server (http://gsds.cbi.pku.edu.cn) (*Hu et al., 2014*) using the coding and genomic sequences of the soybean trihelix family genes. The MEME program (http://meme-suite.org/) (*Bailey et al., 2015*) was used to identify conserved motifs among all the soybean trihelix genes with using default parameters except for the flowering parameter: the maximum number of motifs, 10.

## Gene duplication and Ks calculate

The chromosomal location map was constructed by using the MapChart (v.2.32) program (*Voorrips, 2002*). Major criteria was used for analyzing potential gene duplications; Potential gene duplications were determined by two major criteria: length of aligned sequence covers $\geq$ 75% of longer gene and similarity of aligned regions is $\geq$ 75%. Ka and Ks values were calculated using KaKs Calculator (http://code.google.com/p/kakscalculator/wiki/KaKs_Calculator) (*Zhang et al., 2006*).

## *Cis*-elements analysis of trihelix TF family

The 1,500 bp upstream sequences of 71 soybean trihelix family genes were obtained from Phytozome database and identified as hypothetical promoters. The PLANTCARE database (http://bioinformatics.psb.ugent.be/webtools/plantcare/html/) was used to analyze the *cis*-regulatory elements of the trihelix family gene promoters.

## Mitogen-activated protein kinase (MAPK)-specific phosphorylation sites and miRNA target prediction

MAPK-specific phosphorylation sites were predicted by the Musite (version 1.0) (http://musite.net) tool at the specificity level of $\geq$98% (*Gao et al., 2010*). The online sever psRNA Target (http://plantgrn.noble.org/psRNATarget/, *Dai, Zhuang & Zhao, 2018*) was used to predict miRNA targets with using the parameters as follows: maximum expectation of 3 and maximum energy to unpair target site (UPE) of less than 25.

## Expression analysis of soybean trihelix genes in different tissues using RNA-Seq Atlas

To analyze expression pattern associated with gene function, 71 soybean trihelix family genes were investigated based on the public RNA-seq data published in Phytozome v12.1 (*Libault et al., 2010*; *Goodstein et al., 2011*). The expression of the 71 soybean trihelix family genes were analyzed from 9 different soybean tissues, including flowers, leaves, nodules, pods, roots, root hairs, seeds, shoot apical meristems (SAMs) and stems (*Libault et al., 2010*).

## Analysis of differentially expressed soybean trihelix genes in response to the submergence and MAMP mixture treatment

For the analysis of trihelix genes in response to the submergence stress at the seedling stage, the FPKM (Fragments per kilobase for a million reads) of all the soybean trihelix genes was obtained from our previous transcriptome data of soybean (14-day-old seedlings) at 3 h, 6 h, 12 h and 24 h after submergence (*Lin et al., 2019*). The raw RNA-seq data were deposited in the NCBI Sequence Read Archive (SRA), and the accession number is SRP181976. The soybean trihelix genes that showed more than two-fold expression changes ($|\log2(\text{Fold Change})| \geq 1$) with the adjusted $P$ value of less than 0.05 were considered as differentially expressed genes (DEGs).

For the analysis of trihelix genes in response to the MAMP mixture treatment, publicly available microarray data (series accession no. GSE32642) was obtained from the Gene Expression Omnibus (GEO) database (https://www.ncbi.nlm.nih.gov/geo/), where the soybean leaves were treated by the MAMP mixture (flagellin 22 (flg22) and crab shell chitin) (*Valdés-López et al., 2011*). The microarray data was processed using the NCBI GEO software, GEO2R (https://www.ncbi.nlm.nih.gov/geo/geo2r/). The soybean trihelix genes that showed more than two-fold expression changes ($|\log2(\text{Fold Change})| \geq 1$) with the adjusted $P$ value of less than 0.05 were considered as differentially expressed genes.

## Plant growth conditions and treatments

The seedlings of a soybean cultivar Qihuang 34 were grown in a controlled culture room at 25 °C under long day (16/8 h light/dark) condition. For the submergence treatment, the seedlings (14-day-old) grown in the pots were completely immersed in water by placing them in the white plastic containers filled with water. The whole roots of treated soybean were sampled at 3 h, 6 h, 12 h and 24 h time point and immediately placed in liquid nitrogen and stored at −80 °C until RNA extraction for quantitative real-time PCR (qRT-PCR). All root material from 10 plants was pooled and three biological replicates were performed.

For the NaCl and ABA treatments, 14-day-old seedlings were removed from the soil and cultured in Hoagland liquid medium for 2 days, and then transferred into new Hoagland liquid medium containing 100 mM NaCl or 100 μM ABA, respectively. The whole roots of control and treated seedlings were sampled at 0 h, 3 h, 6 h, and 12 h after treatment. All samples were rapidly frozen in liquid nitrogen and stored at −80 °C. All root material from 5 plants was pooled and three biological replicates were performed.

For the salicylic acid (SA), Jasmonic acid (JA) and ethylene precursor 1-aminocyclopropane-1-carboxylicacid (ACC) treatment, 14-day-old seedlings were

sprayed with 1mM SA, 100 μM JA or 100 μM ACC on the leaf, respectively. The leaves of control and treated seedlings were sampled at 0 h, 30 min, 3 h, 6 h, and 12 h after treatment. All samples were rapidly frozen in liquid nitrogen and stored at −80 °C. Leaves from 5 plants were pooled and three biological replicates were performed.

### qRT-PCR analysis

The total RNA from all of the roots or leaves of soybean plants suffering from submergence/NaCl stress or hormone treatments (ABA, SA, JA and ACC) and the control was isolated using Trizol reagent. qRT-PCR was performed using Roche 480 Light Cycler (Roche, Mannheim, Germany) using a Takara SYBR Premix Extaq (Takara, Japan). Three biological replicates were analyzed, with 3 technical replicates for each of the triplicate biological samples. The calculation of gene expression levels followed the $2^{-\Delta\Delta CT}$ method described by *Livak & Schmittgen (2001)*. *GmActin* (*Glyma.18G290800*) was used as the internal reference gene for the submergence, JA, SA and ACC treatment (*Song et al., 2018*), and *GmELF1b* (*Glyma.02G276600*) was used as the internal reference gene for the salinity and ABA treatments (*Yim et al., 2015*; *Li et al., 2016*). The primers of the selected trihelix genes used for qRT-PCR are listed in Table S9.

## RESULTS

### Identification and duplication analysis of soybean trihelix genes

A total of 71 trihelix genes were identified in soybean genome. In addition to the 63 trihelix genes reported previously, 8 new soybean trihelix genes (*Glyma.04G135400*, *Glyma.05G200400*, *Glyma.08G007800*, *Glyma.10G064900*, *Glyma.15G234100*, *Glyma.17G194700*, *Glyma.20G184500* and *Glyma.20G189000*) were also identified in this study. The characteristics of all these 71 soybean trihelix genes including gene Locus, chromosomes locations, lengths of the CDS (coding sequence) and protein, protein sizes, isoelectric points and subcellular localization were summarized in Table S1. The length of the CDS vary from 753 to 2667 bp, with predicted protein products varying in length from 250 (*Glyma.15G082900*) to 888 (*Glyma.04G194600*) amino acids. Their protein MW (Molecular Weight) range from 28.62 kDa to 97.37 kDa. Their predicted isoelectric points vary from 4.52 (*Glyma.07G087900*) to 10.02 (*Glyma.13G290500*). Almost all of the trihelix proteins were predicted to have nuclear localization except for *Glyma.04G194600*, *Glyma.06G171400* and *Glyma.10G201400* which were localized to the chloroplast (Table S1).

These 71 soybean trihelix genes are distributed on 19 out of 20 chromosomes. Chromosome 10 (Chr 10) contains the largest number of trihelix genes (10 trihelix genes) whereas Chr5 and Chr12 contain only one. No trihelix genes were detected on Chr14 (Fig. 1). We then investigated the gene duplication events of the trihelix genes. No trihelix gene in soybean was identified as a tandem duplication, and 13 pairs of genes were found to be segmental duplications (Fig. 2, Table 1). Simultaneously, the nonsynonymous (Ka) and synonymous (Ks) substitution rates between these duplicated gene pairs were calculated. The Ka/Ks of all the 13 duplicated gene pairs (Table 1) was found to be less than 1, suggesting that the soybean trihelix gene family might have experienced a strong purifying

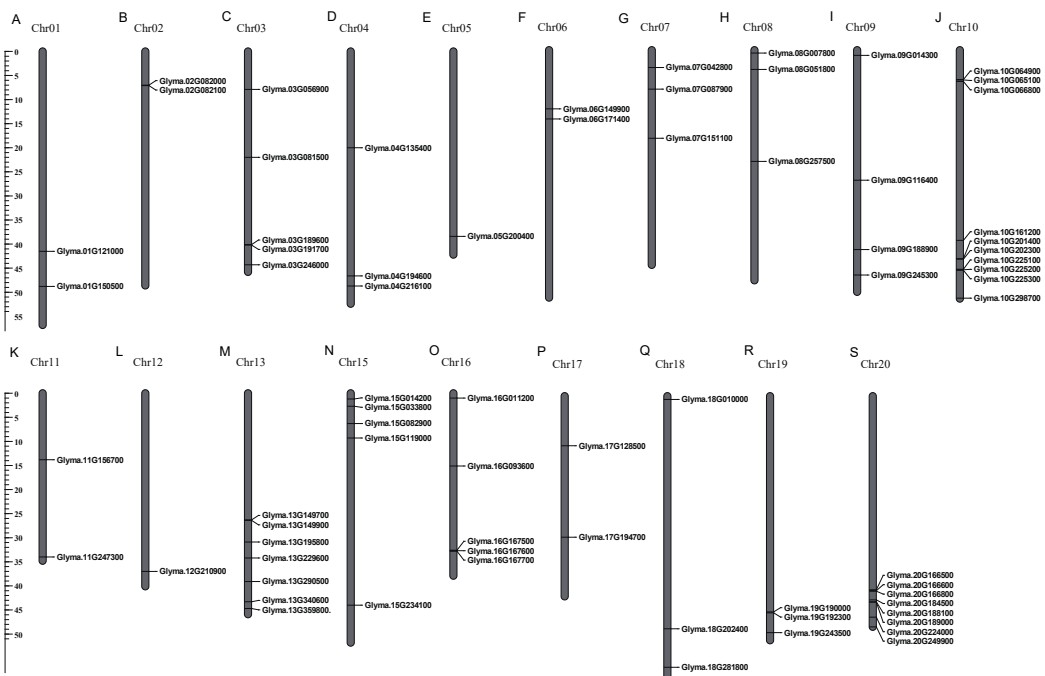

**Figure 1  Chromosomal locations of soybean trihelix genes.** Gray bars represent chromosomes. Chromosome numbers are shown at the tops of the bar. Trihelix genes are labeled at the right of the chromosomes. Scale bar denotes the chromosome lengths (Mb). Chr: Chromosome. A–S: 20 chromosomes.

selective pressure during evolution. Moreover, the approximate date of duplication events was calculated using T = Ks/2λ (*Nei & Sudhir, 2000*), assuming that the divergence rate of 6.161029 synonymous mutations per synonymous site per year for soybean (*Lynch & Conery, 2000*). The segmental duplications of the trihelix genes in soybean originate from 1.67 million years ago (Mya) (Ks = 0.02) to 19.80 Mya (Ks = 0.24) (Table 1), with the mean of 12.18 Mya (Ks = 0.1487).

## Phylogenetic relationships, conserved protein motifs and gene structure analysis of soybean trihelix genes

To uncover the classifications of the trihelix proteins of soybean, a phylogenetic analysis of the amino acid sequences with the trihelix TFs from soybean and other species (Arabidopsis, rice and tomato) was conducted. As shown in Fig. 3, all of the trihelix genes are divided into six subfamilies: SIP1, GTγ, GTδ, GT-1, GT-2 and SH4. This classification is largely similar to the previous analyses of trihelix TFs in Arabidopsis, rice and tomato (*Kaplan-Levy et al., 2012*; *Yu et al., 2015*; *Li et al., 2019*). The 71 soybean trihelix genes are distributed over all of these subfamilies. GT-2 and SIP1 are the largest subfamilies, both containing 22 soybean trihelix genes. Whereas the GTδ subfamily is the smallest, only containing 3 members. This result suggests that soybean trihelix genes are distributed unevenly in the different subfamilies (Fig. 3).

The DNA-binding domain of trihelix genes features a typical trihelix (helix-loop-helix-loop-helix) structure. By performing the multiple sequence alignment, we found that

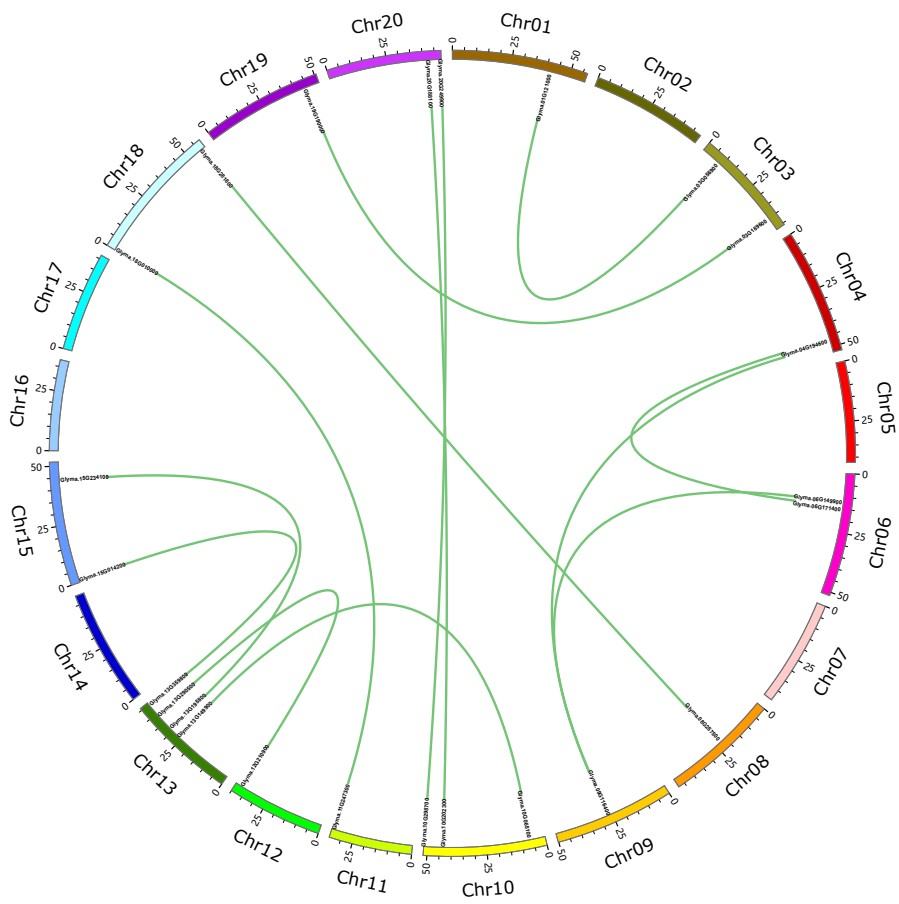

**Figure 2    Circle plot of soybean chromosomes and the 26 trihelix family genes displayed as segmental duplicated gene pairs.** The thick green lines indicate duplicated trihelix gene pairs. Scale bar marked on the chromosome indicates chromosome lengths (Mb). Chr: Chromosome.

GT binding motifs were highly conserved among soybean and Arabidopsis trihelix genes (Fig. S1). Four amino acids of the tress tandem helices were found to be invariant in almost all of the trihelix genes, including tryptophan (W) and leucine (L) in Helix1; tryptophan (W) in Helix2; and cysteine (C) in Helix3 (Fig. S1).

We then used the MEME program to detect conserved motifs in the soybean trihelix family and found 10 distinct motifs (Fig. 4A). Among them, 5 motifs are matched up with the GT binding domains. Motif 2 is in Helix 1 domain. Motif 4 is Helix 1 and Helix 2, motif 10 is Helix 2, motif 1 or motif 6 is Helix 3. The details of these 10 motifs are shown in Fig. S2. In general, genes from the same subfamily are characterized by a similar motif type and distribution. Almost all of the SIP1 subfamily members (except Glyma.15G082900, Glyma.10G065100, Glyma.13G149900, Glyma.10G201400 and Glyma.20G189000) contain Motif 4, 6 and 8. Moreover, the motif 8 is only identified in this subfamily. All of the SH4 subfamily members (expect Glyma.17G194700 and Glyma.20G184500) contain Motifs 2 and 6. GTγ subfamily contains motifs 1 and 7 (except Glyma.10G064900 contains motif 6 and 7). Motif 7 is only present in the GTγ subfamily. Both motif 2 and 10 are present in

**Table 1  Segmental duplications of trihelix paralogous pairs in soybean and inference of duplication time.**

| Trihelix gene pairs | Ka | Ks | Ka/Ks | P-Value (Fisher) | Length (bp) | Approximate duplication date (MYA) |
|---|---|---|---|---|---|---|
| *Glyma.01G121000 &Glyma.03G056900* | 0.0320 | 0.2021 | 0.1584 | 2.22E−11 | 792 | 16.57 |
| *Glyma.04G194600 &Glyma.06G171400* | 0.0156 | 0.1079 | 0.1445 | 2.20E−20 | 2,640 | 8.84 |
| *Glyma.04G216100 &Glyma.09G116400* | 0.0084 | 0.0204 | 0.4141 | 0.041626 | 1,467 | 1.67 |
| *Glyma.08G257500 &Glyma.18G281800* | 0.0234 | 0.1620 | 0.1442 | 1.35E−12 | 972 | 13.28 |
| *Glyma.09G116400 &Glyma.06G149900* | 0.0432 | 0.1596 | 0.2705 | 8.73E−11 | 1,431 | 13.08 |
| *Glyma.10G065100 &Glyma.13G149900* | 0.0176 | 0.1279 | 0.1376 | 8.36E−15 | 1,437 | 10.48 |
| *Glyma.10G202300 &Glyma.20G188100* | 0.0364 | 0.1109 | 0.3286 | 2.45E−07 | 1,593 | 9.09 |
| *Glyma.10G298700 &Glyma.20G249900* | 0.0537 | 0.2249 | 0.2388 | 1.95E−11 | 945 | 18.43 |
| *Glyma.11G247300 &Glyma.18G010000* | 0.0013 | 0.1009 | 0.0126 | 0 | 1,119 | 8.27 |
| *Glyma.12G210900 &Glyma.13G290500* | 0.0252 | 0.2416 | 0.1042 | 1.21E−16 | 909 | 19.80 |
| *Glyma.13G195800 &Glyma.15G234100* | 0.0600 | 0.2071 | 0.2898 | 9.48E−09 | 957 | 16.97 |
| *Glyma.13G359800 &Glyma.15G014200* | 0.0355 | 0.1386 | 0.2565 | 1.71E−07 | 1,146 | 11.36 |
| *Glyma.19G190000 &Glyma.03G189600* | 0.0274 | 0.1288 | 0.2126 | 7.40E−11 | 1,320 | 10.55 |

**Notes.**

Non-synonymous (Ka) and synonymous (Ks) show the substitution rates. Ka/Ks is the ratio of non-synonymous (Ka) versus synonymous (Ks) mutations. This ratio is used as indicator to determine the selective pressure or strength on a protein-encoding gene. Ka/Ks = 1 shows "no selection", Ka/Ks<1 indicates "negative or purifying selection" and Ka/Ks>1 denotes "positive or Darwinian selection".

the GT-1 and GT-2 subfamilies, and the GT-2 subfamily contains more motifs than any other subfamilies. All of the 3 trihelix proteins belonging to GTδ subfamily contain motif 2, 6 and 10 (Fig. 4A).

To identify the features of the soybean trihelix gene family, the structures of these trihelix genes were further analyzed (Fig. 4B). As shown in Fig. 4B, the number of the soybean trihelix gene exons is discontinuously distributed from 1 to 17. The average number of exons is the lowest in GTγ subfamily, and highest in GTδ subfamily (Fig. S3). We found that GTγ genes only had 1–2 exons. 17 out of 22 genes belonging to GT-2 subfamily have 2 exons and only 1 intron. In the SIP1 subfamily, 18 out of 22 members have 1–3 exons, and other 4 members have 7 or 8 exons; members belong to the SH4 subfamily have 2-4 exons. In the GT-1 subfamily, 4 members (*Glyma.01G121000*, *Glyma.07G151100*, *Glyma.18G202400* and *Glyma.01G150500*) have 2 exons, and the other 3 members have 4-5 exons. 3 members (*Glyma.17G128500*, *Glyma.04G194600* and *Glyma.06G171400*) of the GTδ subfamily are the largest genes with 17 exons. In general, members of the same family might share similar gene structure.

## Expression profiling of trihelix genes in different tissues of soybean

To investigate the transcript abundance of soybean trihelix genes, expression profiles covering 9 tissues in soybean were analyzed by utilizing the publicly available RNA-seq dataset published in Phytozome (*Libault et al., 2010*) (Table S2). As shown in Fig. 5, the expression patterns of soybean trihelix genes are generally classified into five groups. Those genes in Group 1 are highly expressed in most of the tested tissues in soybean. The group two genes also display high expression signals but seems to preferentially express in some tissues, such as *Glyma.10G161200* and *Glyma.20G224000* which show markedly higher

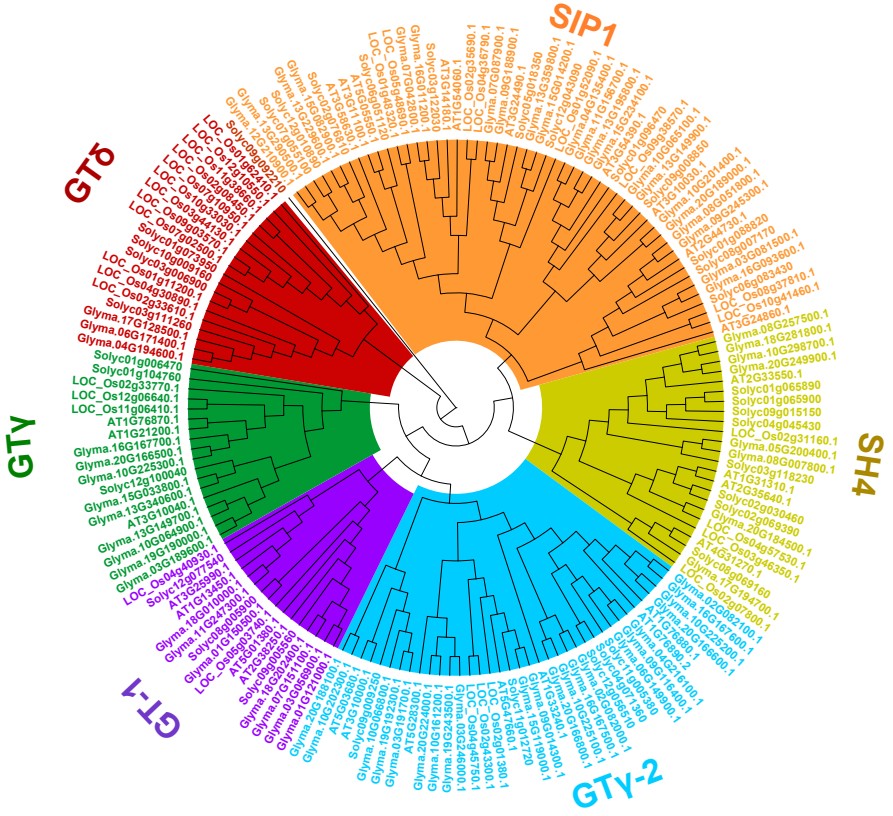

**Figure 3** **Phylogenetic tree based on protein sequences of trihelix genes in soybean, Arabidopsis thaliana, rice and tomato.** Gm, *Glycine max*; AT, *Arabidopsis thaliana*; Os, *Oryza sativa*; Solyc, *Solanum lycopersicon*. The Phylogenetic tree was constructed using the neighbor-joining (NJ) method, with 1,000 bootstrap replicates, using MEGA v.7.0.

transcript abundance profiles in SAM, pods and seeds, but lower expression in roots, nodules and flowers. Group 3 contains 3 genes which are highly expressed in root hairs, nodules and seeds. Genes in Group 4 show low expression levels in almost all of the tissues. It is possible that expression of these genes is induced by a particular condition or that they are pseudogenes. Group 5 contains 30 trihelix genes, of which all of them display relatively low expression levels in most of the tissues, but higher in particular tissues, such as *Glyma.04G135400*, *Glyma.11G156700*, *Glyma.10G202300*, *Glyma.03G191700* and *Glyma.19G192300*, which exhibit low transcript abundance in all of the tested tissues but with high expression levels in stems (Fig. 5).

### *Cis*-elements analysis of soybean trihelix genes

In order to predict the functions of soybean trihelix genes, the *cis*-elements in the promoter regions were analyzed. As shown in Fig. S4, 94.37% of the soybean trihelix genes possess at least one *cis*-element involved in light responsiveness, which is consistent with the light-responsive characteristic of trihelix TFs (*Green, Kay & Chua, 1987*; *Zhou, 1999*). Meanwhile, 3 *cis*-elements related to abiotic stresses were identified, including the anaerobic
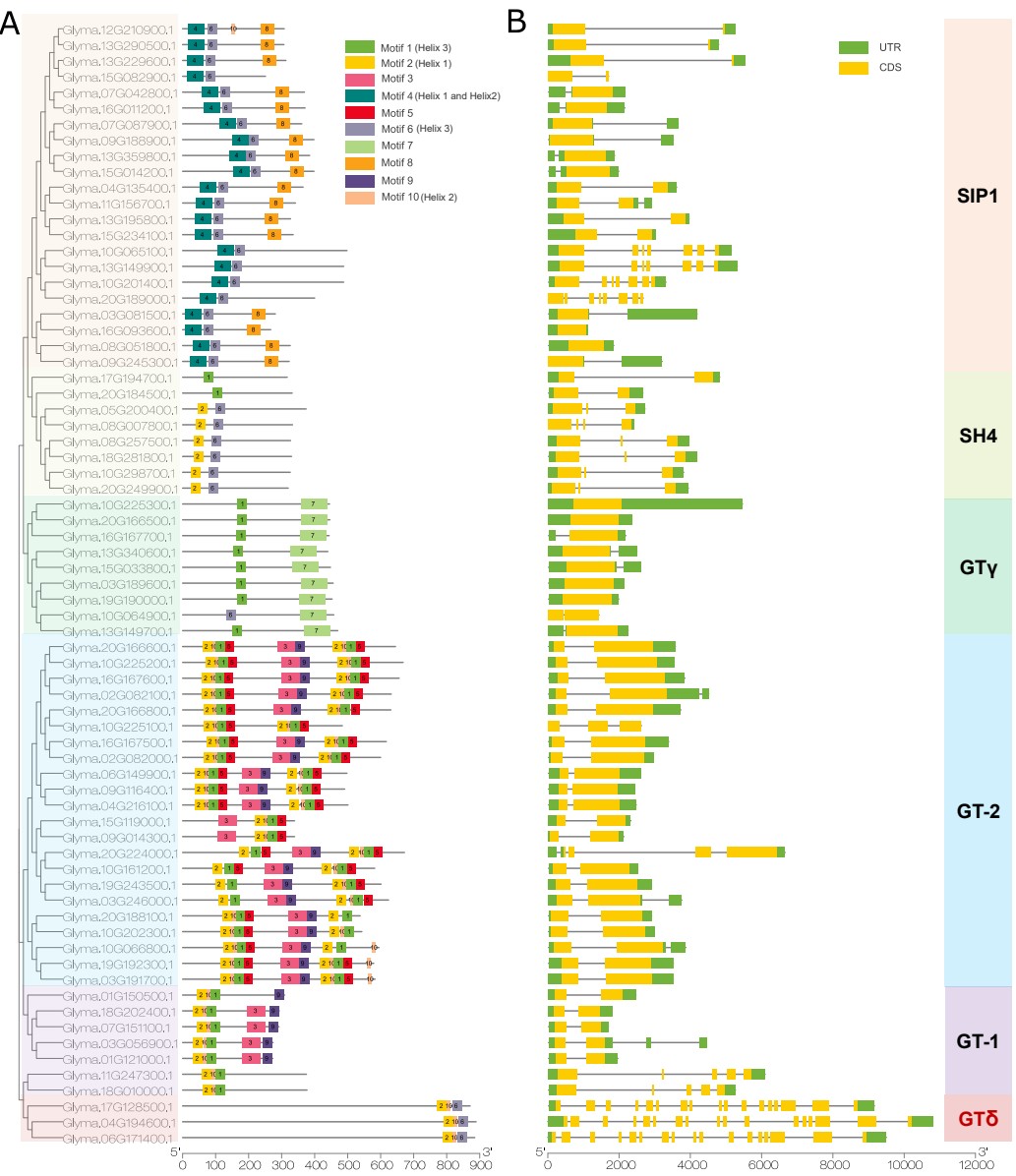

**Figure 4** **The conserved motifs and gene structure of the soybean trihelix family.** The genes in six subfamilies were marked with different colors. (A) The conserved motifs of the soybean trihelix family. The boxes with different colors on the right denote 10 motifs. A detailed motif introduction is shown in Fig. S2. The scale bar at the bottom indicates the lengths of the trihelix protein sequences. (B) Gene structures of the soybean trihelix family. CDS (coding sequence), introns, and untranslated regions (UTRs) are marked by yellow boxes, black lines, and green boxes, respectively. The sizes of exons and introns can be estimated using the scale at the bottom.

responsive element (ARE) (39.44%), drought-responsive elements (DRE and MBS) (19.72%) and low temperature responsive element (LTR) (7.04%). Notably, ARE is much more widely distributed than the other two *cis*-elements related to abiotic stresses (existed in 28 out of 71 (39.4%) trihelix genes) (Fig. S4, Fig. 6). Hormone response elements such as

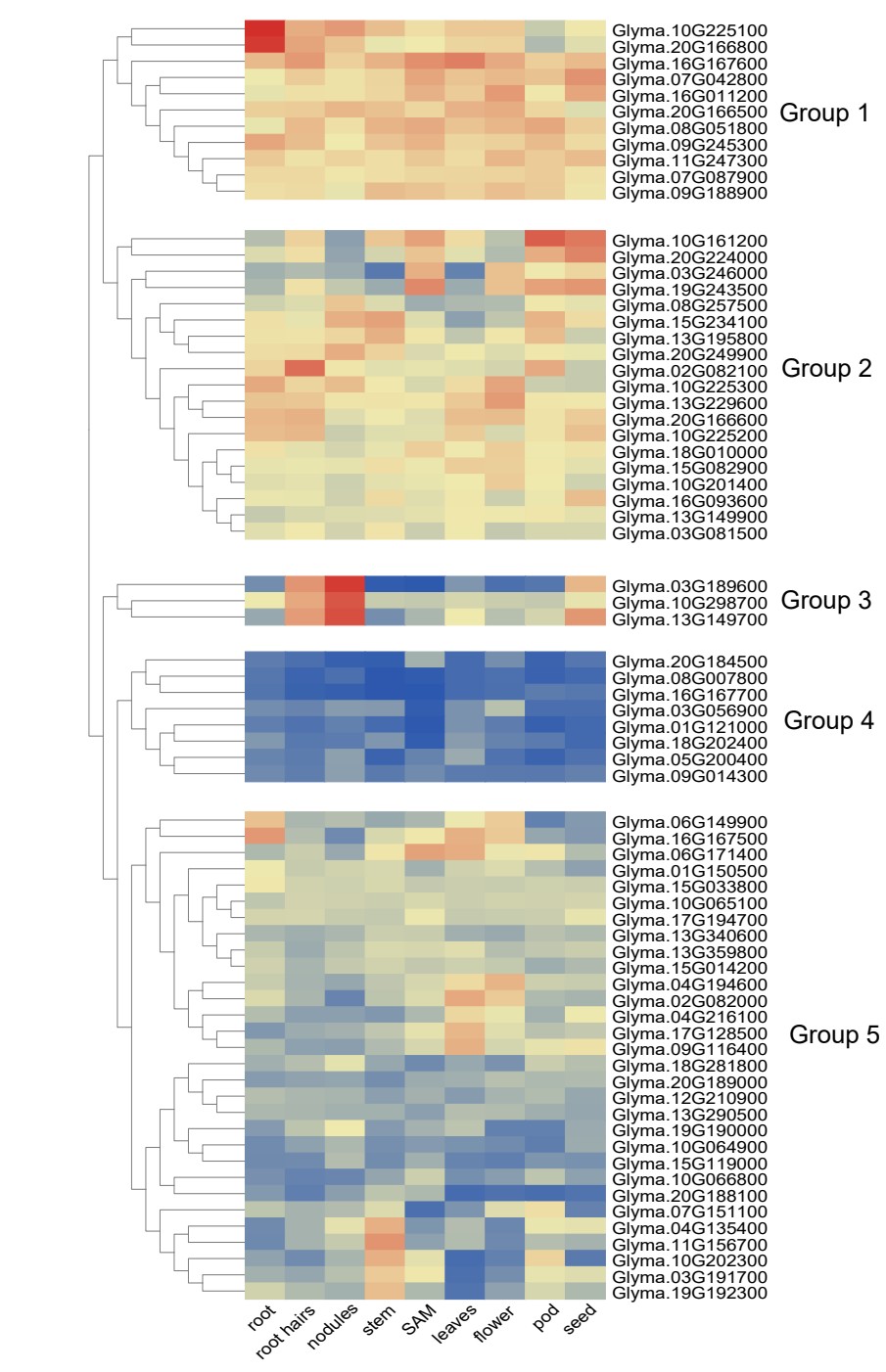

**Figure 5 Heatmap of the expression profiles of the soybean trihelix genes in different tissues.** The expression values (fragments per kilobase for a million reads, FPKM) for each gene in different tissues were log2 transformed before generating heat map. SAM: Shoot Apical Meristem. The expression data was obtained from the public RNA-seq data published in Phytozome v12.1 (*Libault et al., 2010*; *Goodstein et al., 2011*).

abscisic acid responsive element (ABRE) (43.66%), methyl jasmonate responsive element (TGACG-motif) (47.89%), ethylene-responsive element (ERE) (59.15%) and salicylic acid responsive element (TCA-element) (14.08%) were also analyzed and displayed in Fig. 6 and Table S3. All of the 71 trihelix genes (except *Glyma.01G150500*) possess at least 1 stress-responsive *cis*-element (Fig. 6). These results suggest that trihelix genes may be involved in responses to abiotic and biotic stresses.

## Mitogen-activated protein kinase (MAPK)-specific phosphorylation sites and miRNA target prediction

It was reported that MAMP (microbe-associated molecular patterns) treatments induced rapid phosphorylation of an Arabidopsis trihelix gene, *ASR3*, via MAP KINASE4 (*Li et al., 2015*). This fact suggests that trihelix genes could be post-translationally regulated by phosphorylation. Therefore, we also predicted the presence of putative MAPK phosphorylation sites. 29 soybean trihelix genes which are distributed in all of the 6 subfamilies have at least one putative MAPK phosphorylation site. Among them, 22 (75.9%) belong to the GT-2 subfamily (containing 9 trihelix genes have at least one MAPK phosphorylation site) and SIP1 subfamily (containing 13 trihelix genes have at least one MAPK phosphorylation site). In the GT-2 subfamily, the putative MAPK phosphorylation sites are mostly threonine residue. While in the SIP1 subfamily, the putative MAPK phosphorylation sites are mostly serine residues (Table S4).

MicroRNAs (miRNAs) are a class of small non-coding regulatory RNAs that regulate gene expression by guiding target mRNA cleavage or translational inhibition (*Unver, Namuth-Covert & Budak, 2009*; *Eldem, Okay & Unver, 2013*; *Zhang, 2015*). In order to identify the involvement of miRNAs in regulating the expression of soybean trihelix genes, putative miRNA targets were also analyzed. As shown in Table S5, 26 soybean trihelix genes were predicted to be targeted by 27 miRNAs, with inhibition of cleavage or translation. *Glyma.11G247300*, *Glyma.19G243500* and *Glyma.17G128500* contain three target sites; *Glyma.18G010000*, *Glyma.07G151100*, *Glyma.02G082100* and *Glyma.10G225300* contain two target sites; the other 19 trihelix genes have only one target site. Additionally, both of the gma-miR1533 and gma-miR1512a-3p can target 3 of the trihelix genes.

## The expression profile of soybean trihelix genes in response to environmental stresses

Previous studies showed that the maize trihelix genes could respond to flood stress (*Du, Huang & Liu, 2016*). Considering that many soybean trihelix genes possess the anaerobic responsive element (ARE) in their promoters, we analyzed our previous transcriptome data of the soybean cultivar Qihuang34 submerged for 3, 6, 12 and 24 h at the seedling stage (*Lin et al., 2019*) (Table S6). 45 trihelix genes display more than 2-fold expression level differences compared with controls in at least one time point (Fig. 7, Table S7). 25, 25, 37 and 31 differentially expressed genes (DEGs) were identified after submergence treatment at 3, 6, 12 and 24 h, respectively (Table S7). 15 soybean trihelix genes are significantly changed at all of the 4 time points (Table 2). These 15 DEGs are distributed in 5 subfamilies including SIP1, SH4, GTγ, GT-1 and GT-2 (Table 2). Among these 15 DEGs, 4 genes (*Glyma.11G156700, Glyma.20G224000, Glyma.05G200400,* and

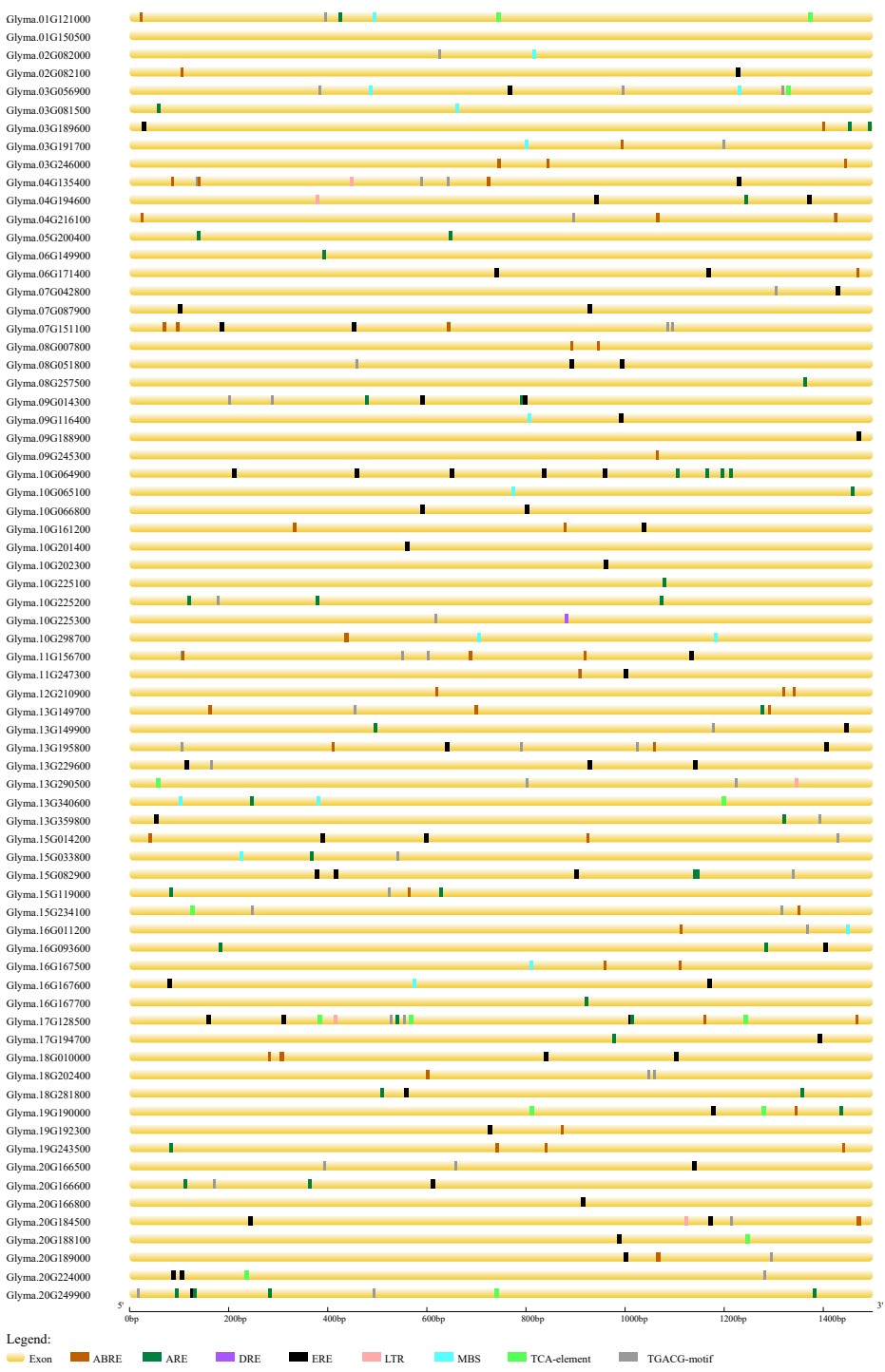

**Figure 6** Predicted *cis*-elements in the promoter regions (1,500 bp upstream from the transcription initiation site) of the soybean trihelix genes. The scale bar at the bottom indicates the length of promoter sequence. ABRE: abscisic acid responsive element; ARE: anaerobic responsive element; DRE: drought responsive element; ERE: ethylene responsive element; LTR: low temperature responsive element; MBS: drought responsive element; TCA-element: salicylic acid responsive element; TGACG-motif: methyl jasomonate responsive element.

*Glyma.08G007800*) are down-regulated, and the remaining genes are all up-regulated. We then performed qRT-PCR assays of these 15 DEGs with the root samples from the Qihuang 34 plants suffered submergence (Fig. S5). As shown in Fig. S5, the expression of 11 genes (*Glyma.03G189600*, *Glyma.06G149900*, *Glyma.07G042800*, *Glyma.07G151100*, *Glyma.08G007800*, *Glyma.09G116400*, *Glyma.11G247300*, *Glyma.13G149700*, *Glyma.13G229600*, *Glyma.15G082900* and *Glyma.19G19000*) is significantly changed under any timepoint, which is consistent with our previous RNA-Seq results. Notably, 3 GTγ subfamily genes (*Glyma.03G189600*, *Glyma.13G149700* and *Glyma.19G190000*) are all up-regulated, and represent the most significant change (Table 2, Fig. S5). The other 4 genes (*Glyma.11G157600*, *Glyma.16G167600*, *Glyma.20G224000* and *Glyma.05G200400*) are only up- or down-regulated at specific time points after flood treatment (Fig. S5).

To gain more insights into the role of trihelix genes in other abiotic stress responses, we performed qRT-PCR assays to examine the expression profiles of 12 selected trihelix genes under high-salt stress and ABA treatment (Fig. 8). These 12 genes includes the 3 GTγ subfamily genes *(Glyma.03G189600*, *Glyma.13G149700* and *Glyma.19G190000*) which represent the most significant change under flood treatment and other 9 randomly selected trihelix genes. According to the results, the expression of 6 genes is changed when plants were treated with NaCl (Figs. 8A–8L). 3 genes (*Glyma.09G116400*, *Glyma.13G195800* and *Glyma.20G166800*) are down-regulated by NaCl treatment, while 3 genes (*Glyma.03G189600*, *Glyma.13G149700* and *Glyma.19G190000*) are up-regulated 6 h after NaCl treatment. The other 6 genes didn't show obvious changes (Figs. 8A–8L). As for the ABA treatment (Figs. 8M–8X), 1 trihelix gene (*Glyma.07G151100*) didn't show obvious change compared to the control, while the other 11 tested trihelix genes were up- or down-regulated under ABA treatment (Figs. 8M–8X). Among them, *Glyma.13G195800* is down regulated 3 h after ABA treatment, and the other 10 genes are up-regulated at different time point (Figs. 8M–8X). It is noteworthy that the 3 GTγ genes (*Glyma.03G189600*, *Glyma.13G149700* and *Glyma.19G190000*) which are extremely sensitive to flooding are all significantly up-regulated by NaCl treatment and ABA treatment at an exact time point (Table 2, Fig. 8).

It has been shown that trihelix genes may be involved in plant defense responses. Microbe-associated molecular pattern (MAMP)-triggered immunity (MTI) is an important component of the plant innate immunity response to invading pathogens (*Kim et al., 2011*). In the current study, we used the publicly available microarray data (series accession no. GSE32642) to analyze the expression of trihelix genes in soybean leaves which were treated by the MAMP mixture (flg22 and chitin) (*Valdés-López et al., 2011*). This analysis showed the expression of 2 soybean trihelix genes were significantly changed 30 min after the MAMP mixture treatment (Fig. 9A, Table S8). *Glyma.01G121000* is significantly up-regulated, while *Glyma.10G066800* is down-regulated. Phytohormones, such as JA, SA and ethylene regulate plant defenses against diverse pathogens. We next examined the expression pattern of these 2 trihelix genes in soybean leaves under JA, SA and ACC treatment by qRT-PCR. As shown in Fig. 9, both of these two genes are sensitive to SA, JA and ACC. *Glyma.01G121000* is up-regulated under all of these treatments, while *Glyma.10G066800* displays a more complex expression pattern. Under SA treatment,

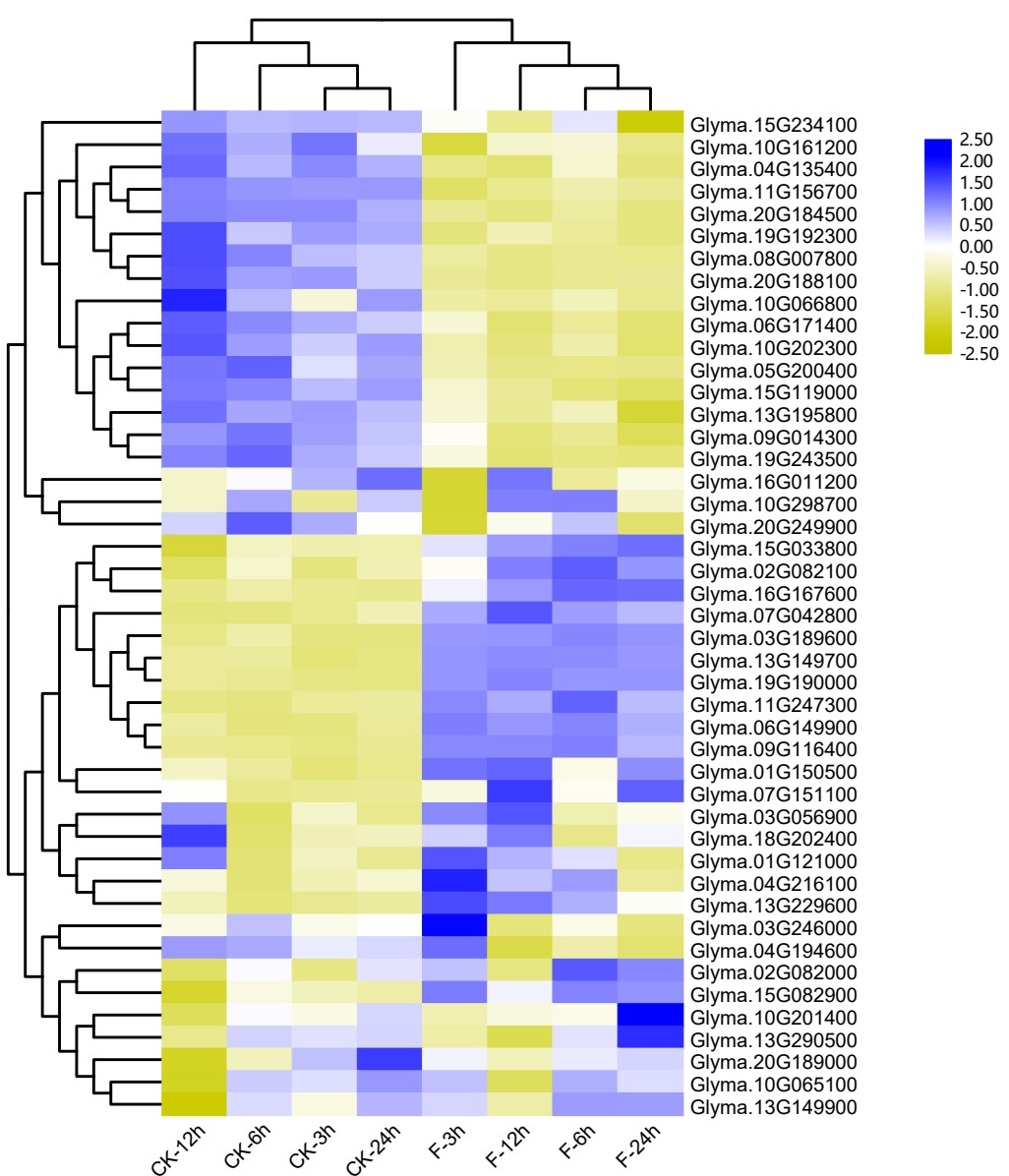

**Figure 7** **Heatmap of the differentially expressed trihelix genes of soybean suffering from submergence treatment.** The RNA-seq data was obtained from NCBI Sequence Read Archive (SRA) with the accession number: SRP181976 (*Lin et al., 2019*). The expression values (Fragments per kilobase for a million reads, FPKM) for each gene were log2 transformed before generating the heatmap. ck-3 h, ck-6 h, ck-12 h and ck-24 h indicate the roots from plants untreated by submergence stress; F-3 h, F-6 h, F-12 h, F-24 h denote the roots sampled at 3 h, 6 h, 12 h and 24 h after submergence stress treatment, respectively.

*Glyma.10G066800* is up-regulated 30 min and 12 h after treatment, but is down-regulated 3 h after treatment. Under JA treatment, *Glyma.10G066800* is up-regulated 30 min after treatment, but is down-regulated 3 and 12 h after treatment. Under ACC treatment, it is down regulated at 3 h and 12 h after treatment.

**Table 2** The differently expressed soybean trihelix genes at 4 time points after submergence treatment identified by analyzing the public-RNA seq data.

| Gene ID | clade | log2FC (3 h) | adjusted p value (3 h) | log2FC (6 h) | adjusted p value (6 h) | log2FC (12 h) | adjusted p value (12 h) | log2FC (24 h) | adjusted p value (24 h) |
|---|---|---|---|---|---|---|---|---|---|
| *Glyma.13G229600* | SIP1 | 1.78 | 1.61E−30 | 1.59 | 7.00E−21 | 1.64 | 2.48E−09 | 1.14 | 7.24E−14 |
| *Glyma.11G156700* | SIP1 | −1.93 | 4.68E−11 | −1.04 | 3.34E−04 | −1.46 | 9.01E−04 | −1.26 | 1.10E−05 |
| *Glyma.07G042800* | SIP1 | 1.69 | 9.01E−33 | 1.94 | 7.20E−41 | 2.43 | 2.10E−24 | 1.54 | 1.47E−16 |
| *Glyma.15G082900* | SIP1 | 1.31 | 2.51E−17 | 1.26 | 3.32E−14 | 1.6 | 5.60E−09 | 1.47 | 7.56E−12 |
| *Glyma.03G189600* | GTγ | 10.27 | 8.06E−161 | 7.83 | 8.13E−49 | 9.76 | 1.56E−286 | 10.87 | 1.04E−142 |
| *Glyma.13G149700* | GTγ | 7.99 | 6.01E−278 | 6.78 | 2.31E−67 | 7.12 | 3.91E−174 | 7.68 | 9.37E−287 |
| *Glyma.19G190000* | GTγ | 8.66 | 4.32E−232 | 8.15 | 1.89E−83 | 8.57 | 0.00E+00 | 8.94 | 6.04E−146 |
| *Glyma.07G151100* | GT-1 | 1.03 | 4.18E−03 | 1.29 | 3.58E−06 | 1.8 | 9.12E−06 | 2.14 | 1.08E−12 |
| *Glyma.11G247300* | GT-1 | 1.53 | 7.30E−16 | 1.91 | 5.00E−19 | 1.68 | 8.41E−14 | 1.45 | 1.39E−10 |
| *Glyma.16G167600* | GT-2 | 1.29 | 2.63E−11 | 2.02 | 1.35E−23 | 2.04 | 2.25E−14 | 2.23 | 2.32E−19 |
| *Glyma.06G149900* | GT-2 | 3.14 | 2.06E−59 | 3.23 | 6.49E−105 | 2.73 | 2.54E−19 | 2.54 | 4.74E−23 |
| *Glyma.20G224000* | GT-2 | −2.19 | 2.22E−05 | −1.5 | 2.54E−03 | −1.67 | 2.92E−02 | −2.06 | 4.93E−05 |
| *Glyma.09G116400* | GT-2 | 4.04 | 7.48E−90 | 4.06 | 5.68E−107 | 3.94 | 2.36E−38 | 3.32 | 6.56E−40 |
| *Glyma.05G200400* | SH4 | −1.43 | 1.33E−02 | −5.29 | 2.74E−08 | −5.72 | 1.11E−09 | −5.36 | 2.72E−08 |
| *Glyma.08G007800* | SH4 | −1.92 | 2.31E−02 | −2.72 | 5.30E−03 | −4.45 | 8.90E−06 | −2.74 | 6.54E−03 |

**Notes.**
The genes written in red and black color were up-and down-regulated, respectively. The RNA-seq data was obtained from NCBI Sequence Read Archive (SRA) (SRP181976) (*Lin et al., 2019*). Log2FC, Log2FoldChange.

## DISCUSSION

### Characterization of the soybean trihelix genes

In this study, we identified 71 soybean trihelix genes by the Myb/SANT-LIKE domain using an HMM (Hidden Markow Model)- search. We obtained more trihelix genes in the present study than in the previous report (*Osorio et al., 2012*) because the trihelix genes were identified in the new version of the soybean genome database Wm82.a2. v1 and performing a different search method. To date, the soybean trihelix family is the second largest known trihelix gene family (next to the wheat trihelix family composed of 94 trihelix genes) (*Xiao et al., 2019*) compared with other plants, such as Arabidopsis (28), rice (41) tomato (36), *Brassica rapa* (52), Chrysanthemum (20) and tartary buckwheat (31) (*Yasmeen et al., 2016*; *Yu et al., 2015*; *Song et al., 2016*; *Wang et al., 2017*; *Wang et al., 2018*; *Li et al., 2019*; *Ma et al., 2019*).

Gene duplication is one of the major evolutionary mechanisms for generating novel genes that help organisms adapt to different environments (*Moore & Purugganan, 2003*; *Kong et al., 2007*). In general, gene families expand mainly by tandem and segmental duplications (*Kong et al., 2007*). Only 13 pairs of segmental duplicated genes (36.6%) were identified in soybean among total 71 trihelix genes (Fig. 2, Table 1). This finding suggests that more than half of the trihelix genes in soybean may not originate from the same ancestor. A similar phenomenon was reported in the rice trihelix gene family: it contained only six pairs of duplicated genes among a total of 41 (29.3%) rice trihelix genes (*Li et al., 2019*). The Soybean genome has undergone two rounds of whole genome duplication events at

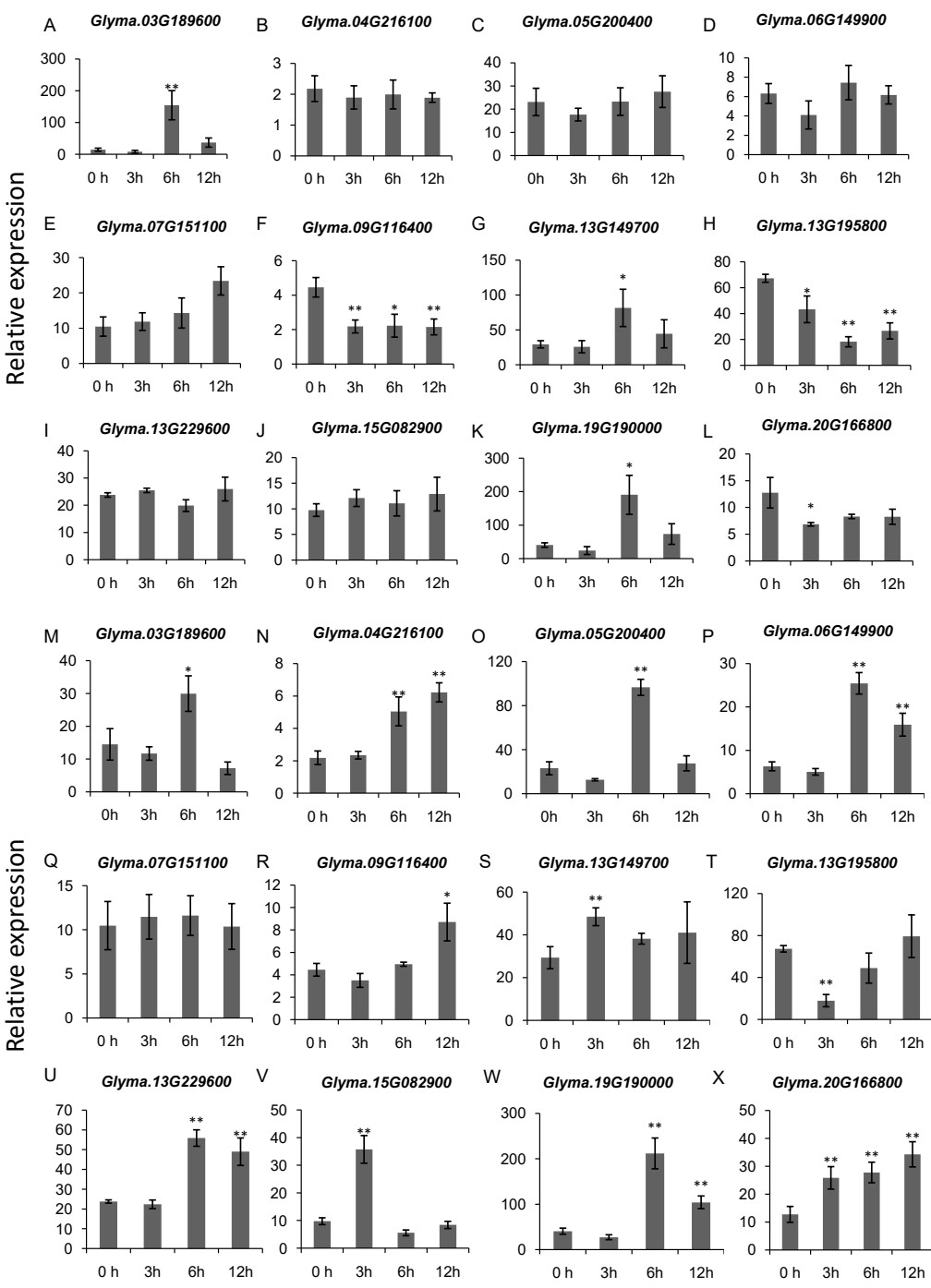

**Figure 8** **The expression profiles of 12 selected trihelix genes under NaCl and ABA treatment examined by qRT-PCR.** (A–L) The relative expression levels of 12 selected trihelix genes under NaCl treatment. 0 h, 3 h, 6 h, and 12 h denoted the roots sampled 0 h, 3 h, 6 h, and 12 h after 100 mM NaCl treatment, respectively. The data represent the mean ± SD of three biological replicates. The statistical significance was determined using Student's $t$-tests (**$p < 0.01$, *$p < 0.05$, $n = 3$); (M–X) The relative expression levels of 12 selected trihelix genes under ABA treatment. 0 h, 3 h, 6 h, and 12 h denoted the roots sampled 0 h, 3 h, 6 h, and 12 h after 100 μM ABA treatment, respectively. The data represent the mean ± SD of three biological replicates. The statistical significance was determined using Student's $t$-tests (**$p < 0.01$, *$P < 0.05$, $n = 3$).

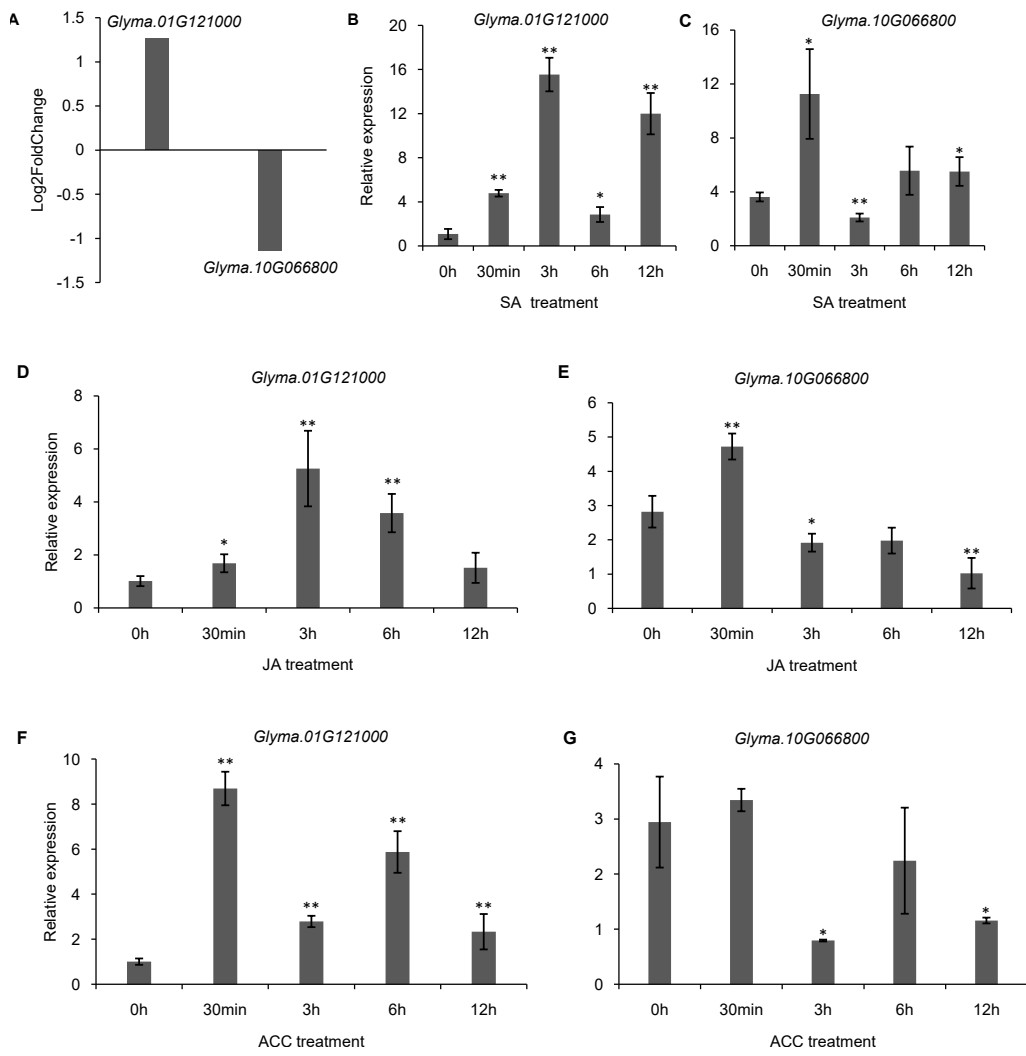

**Figure 9** **The DEGs (differentially expressed genes) of soybean trihelix genes under MAMP (microbe-associated molecular patterns) mixtures treatment and their expression patterns under SA, JA and ACC treatment.** (A) Two DEGs (*Glyma.01G121000* and *Glyma.10G066800*) in soybean leaves under MAMP (mixture with flg22 and chintin) treatment for 30 min. The microarray data was obtained from GEO series accession number GSE32642 (*Valdés-López et al., 2011*). (B–C) The expression pattern of *Glyma.01G121000* and *Glyma.10G066800* in soybean leaves under SA treatment examined by qRT-PCR. (D–E) The expression pattern of *Glyma.01G121000* and *Glyma.10G066800* in soybean leaves under JA treatment examined by qRT-PCR; (F–G) The expression pattern of *Glyma.01G121000* and *Glyma.10G066800* in soybean leaves under ACC treatment examined by qRT-PCR. The data represent the mean ± SD of three biological replicates. The statistical significance was determined using Student's $t$-tests (**$p < 0.01$, *$p < 0.05$, $n = 3$). SA: salicylates; JA: Jasmonic acid; ACC: ethylene precursor 1-aminocyclopropane-1-carboxylicacid.

approximately 59 and 13 million years ago (*Schmutz et al., 2010*). In the present study, all of the segmental duplication events in the soybean gene trihelix family occurred during the recent whole genome duplication event when glycine-specific duplication occurred (Table 1). The duplicated genes which experience subfunctionalization through purifying

selective pressure (Ka/Ks <1) (*Cusack & Wolfe, 2007*), neofunctionalization through positive selective pressure (Ka/Ks >1) (*Blanc & Wolfe, 2004*) and nonfunctionalization play important roles in adaptive evolution (*Flagel & Wendel, 2009*). In this study, all of the Ka/Ks ratios in different trihelix gene pairs are less than 1 (Table 1), suggesting that all of those duplicated genes experience a strong purifying selective pressure and subfunctionalized during evolution. This phenomenon is consistent with the fact that most duplicated genes are subfunctionalized in soybean (*Roulin et al., 2013*).

Trihelix family genes had been classified into three distinctive subfamilies (GTα, GTβ, and GTγ) previously (*Fang et al., 2010*). Then, *Kaplan-Levy et al. (2012)* classified trihelix genes from rice (*Oryza sativa*) and Arabidopsis into five clades, named GT-1, GT-2, SH4, SIP1, and GTγ. Recently, a new subfamily, GTδ was formed in tomato (*Solanum lycopersicum*) and rice (*Yu et al., 2015*; *Li et al., 2019*). In the present study, the classification of the soybean trihelix gene family has been renewed according to their new classification in Arabidopsis, tomato and rice (Fig. 3) (*Kaplan-Levy et al., 2012*; *Yasmeen et al., 2016*; *Yu et al., 2015*; *Li et al., 2019*). All of the 71 soybean trihelix genes were classified into 6 subfamilies (GT-1, GT-2, SH4, SIP1, GTγ and GTδ). The 3 genes in the GTδ subfamily have a large number of exons (17 exons), depicting a significant difference in gene structure from the other subfamily genes (Fig. 4). This finding is similar to *SlGT-12*, a member of the GTδ subfamily in tomato (*Yu et al., 2015*). In addition, 68 soybean trihelix genes were predicted to localize to the nucleus, and only 3 were predicted to localize to the chloroplast (Table S1). We noticed that two of those chloroplast localized genes were from the GTδ-subfamily. Therefore, the members in the GTδ subfamily might serve very different functions from trihelix genes in other subfamilies. Although most of the trihelix genes in soybean may not originate from the same ancestor, the type and number of motifs of soybean trihelix proteins were generally similar within the same sub-family (Fig. 4). Furthermore, members in the same sub-family generally share similar structures and exon numbers (Fig. 4). These results suggest that the trihelix genes belonging to the same subfamily might play similar roles in soybean development, and the conserved motifs may play roles in subfamily-specific functions.

## Expression analysis of soybean trihelix genes

Expression analyses could provide insight into the potential functions of genes. Therefore, the expression pattern of 71 trihelix genes in different tissues of soybean was performed using publicly available RNA-seq data. Many tested trihelix genes were broadly expressed, while there was also a high number of trihelix genes exhibiting specific expression patterns (Fig. 5). Among the 13 pairs of duplicate genes, more than half of the duplicate gene pairs showed similar expression between the duplicates (Fig. S6). For example, *Glyma.04G216100* and its paralog *Glyma.09G116400* had higher expression in leaves. *Glyma.13G195800* and its paralog *Glyma.15G234100* had higher expression in nodules, stem and pod, suggesting that they might have redundant roles. 5 duplicated genes pairs seemed to have different expression patterns. For example, *Glyma.09G116400* was highly expressed in leaves, while its paralog *Glyma.06G149900* had higher expression in root and flower. *Glyma.04G194600* had a relative higher expression level in flower, while its paralog *Glyma.06G171400* had

higher expression in SAM and leaves. This finding suggests that they might have different functions.

Nowadays, various areas in the world face abiotic stresses, such as flooding and salinity. The involvement of the stress responsive genes in various metabolic processes contributes to enhancing stress tolerance in plants. It has been reported that trihelix genes could respond to the waterlogging and salt stress (*Fang et al., 2010*; *Xie et al., 2009*; *Xi et al., 2012*; *Giuntoli et al., 2014*; *Luo et al., 2017*; *Du, Huang & Liu, 2016*; *Magwanga et al., 2019*). Lack of oxygen has been proposed as the main problem associated with flooding (*Drew, 1983*; *Drew, 1997*). In this study, 39.4% of the soybean trihelix genes contain at least one anaerobic responsive element (ARE) in their promoters (Fig. 6, Table S3), implying that the soybean trihelix genes might be involved in flood response. 11 trihelix genes that respond to flood stress at all of the 4 time points were identified in this study (Fig. 7, Table 2, Fig. S5), and all of these genes contained at least 1 stress-responsive *cis*-element (Fig. 6). Among them, 3 genes (*Glyma.03G189600*, *Glyma.13G149700* and *Glyma.19G190000*) which belong to the GTγ subfamily showed the highest sensitivity to the submergence treatment (Table 2). Group VII ethylene response factors (ERF-VIIs) are a class of ERF TFs that regulate the expression of a wide range of genes involved in adaptive responses to flooding and low oxygen levels (*Voesenek & Bailey-Serres, 2015*). It has been reported that *HRA1* (*At3G10040*), a submergence or hypoxia-inducible GTγ trihelix gene in Arabidopsis, acted as a negative regulator of an ERF-VII TF RAP2.12, and then reducing the expression of core hypoxia-response genes (*Giuntoli et al., 2014*). This regulatory mechanism may contribute to the avoidance of RAP2.12 overaccumulation, preventing rapid depletion of carbohydrate reserves under oxygen deprivation (such as during submergence) (*Fukao et al., 2019*). In the present study, of the 3 GTγ genes mentioned above, the expression level of *Glyma.19G190000*, which is homologous to *HRA1* was also strongly induced by submergence treatment, suggesting its potential role in flood tolerance. Until now, none of the flood-responsive trihelix genes have been characterized in soybean. The trihelix genes which were continuously induced by submergence treatment, specifically *Glyma.19G190000* could be candidate genes for further study on flood tolerance in soybean.

In addition, our study also found that certain trihelix genes could respond to high-salt stress (Figs. 8A–8L). Previously, a soybean trihelix gene, *GmGT-2A* (*Glyma.04G216100*), was reported to be up-regulated by NaCl. Moreover, overexpression of the *GmGT-2A* improved plant tolerance to salt (*Xie et al., 2009*). In our study, no significant change in its expression level was detected under NaCl treatment (Fig. 8B). This different expression pattern might result from the different external NaCl concentration (*Xie et al.* used 150 mM NaCl but we used 100 mM NaCl in this study). Perhaps the lower NaCl concentration used in this study was not high enough to induce its expression. In rice, 3 trihelix genes belonging to the GTγ subfamily (*OsGTγ-1*, *OsGTγ-2* and *OsGT γ-3*) were strongly induced by abiotic stress such as drought, cold and salt. In addition, overexpression of *OsGTγ-1* in rice enhanced salt tolerance at the seedling stage, suggesting that the OsGTγ subfamily may participate in the regulation of stress tolerance in rice (*Fang et al., 2010*). In our study, we found that the 3 GTγ genes (*Glyma.03G189600*, *Glyma.13G149700* and *Glyma.19G190000*) which were showing the highest sensitivity to submergence were also

induced by NaCl. This finding suggests that the GTγ subfamily genes in soybean may have potential functions in the stress tolerance as in rice.

Abscisic acid (ABA), which is commonly referred to a stress hormone, is involved not only in regulating stomatal opening, growth and development but also in abiotic stress responses in plants (*Tuteja, 2007*). Some trihelix genes were found to respond to ABA treatment (*Xie et al., 2009*; *Fang et al., 2010*; *Xi et al., 2012*). Notably, the *BnSIP1-1* gene from *Brassica napus* was reported to play roles in ABA synthesis and signaling, salt and osmotic stress response (*Luo et al., 2017*). In this study, among the 71 trihelix genes in soybean, 31 genes contain ABA-responsive elements (Fig. 6, Table S3). By performing qRT-PCR, all of the selected genes (except *Glyma.07G151100*) were found to indeed be regulated by ABA treatment. We focused on the expression of the 3 GTγ genes mentioned above and found that all of them could respond to ABA treatment at a specific time point. This result was in concordance with the fact that all of them possess ABA-responsive elements (Fig. 6, Table S3). Taken together, the 3 GTγ genes of the trihelix family could be the best candidates for further study of abiotic stress tolerance in soybean.

Microbe-associated molecular pattern (MAMP)-triggered immunity (MTI) is an important component of the plant innate immunity and responses to invading pathogens (*Kim et al., 2011*). In Arabidopsis, trihelix TF GT2-like 1 (GTL1) plays an important role in coordinating plant immunity (*Völz et al., 2018*). Moreover, ASR3 functions as a transcriptional repressor regulated by MAMP-activated MPK4 to fine-tune plant immune gene expression (*Li et al., 2015*). In the present study, *Glyma.01G121000* was induced by MAMP mixture, JA, SA and ACC treatments, suggesting that it might also be involved in coordinating plant immunity in soybean. This initial research of soybean trihelix genes should be followed by further work that focuses on plant transformation and phenotyping in order to thoroughly explore trihelix functions in abiotic and biotic stress responses, and the crosstalk between this gene family and hormone signals.

## CONCLUSIONS

In conclusion, 71 members of the trihelix gene family were identified in the newest version of the soybean genome and were classified into 6 subfamilies based on phylogenetic relationships. Genes in the same subfamilies generally share similar gene structure and conserved functional domains. 13 segmental duplicated gene pairs were identified and all of them experienced a strong purifying selective pressure during evolution. By investigating the differential expression profiles of the trihelix genes under flooding, high-salt stresses and MAMP treatment, several candidates for further study of soybean stress-tolerance were identified. These results provide a basis to elucidate the function and the molecular mechanism of trihelix genes on plant development and stress tolerance.

## ACKNOWLEDGEMENTS

We thank Dr. Zhanji Liu for his assistance on bioinformatics analysis during the manuscript preparation.

### Funding

This work was supported by the Agricultural Scientific and Technological Innovation Project of Shandong Academy of Agricultural Sciences (CXGC2018E01) and China Agriculture Research System (CARS-04-CES16). The funders had no role in study design, data collection and analysis, decision to publish, or preparation of the manuscript.

### Grant Disclosures

The following grant information was disclosed by the authors:
Agricultural Scientific and Technological Innovation Project of Shandong Academy of Agricultural Sciences: CXGC2018E01.
China Agriculture Research System: CARS-04-CES16.

### Competing Interests

The authors declare there are no competing interests.

### Author Contributions

- Wei Liu conceived and designed the experiments, performed the experiments, prepared figures and/or tables, authored or reviewed drafts of the paper, and approved the final draft.
- Yanwei Zhang, Wei Li nd Yanhui Lin performed the experiments, prepared figures and/or tables, and approved the final draft.
- Caijie Wang and Ran Xu analyzed the data, prepared figures and/or tables, and approved the final draft.
- Lifeng Zhang conceived and designed the experiments, analyzed the data, authored or reviewed drafts of the paper, and approved the final draft.

### Data Availability

    Data is available at NCBI Sequence Read Archive (SRA, accession numbers: SRP181976).

### Supplemental Information

Supplemental information for this article can be found online at http://dx.doi.org/10.7717/peerj.8753#supplemental-information.

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
