# Peer review of "Genome-wide characterization and expression analysis of soybean trihelix gene family"

_PeerJ, doi:10.7717/peerj.8753_

## Round 0.1 · original submission · Major Revisions

Dear Dr. Zhang,

The reviewers find the paper interesting, but they highlight some deficiencies. Particularly, according to the reviewers some parts lack a clear biological explanation. Moreover, the writing frequently does not conform to correct scientific style and it is grammatically incorrect. For this reason the English language have to be revised and considerably improved.

I encourage you to improve the manuscript according to tips of reviewers. Please, respond point-to-point to the comments of reviewers.

Once again, thank you for submitting your manuscript to PeerJ and we look forward to receiving your revision.

Sincerely,
Gabriele Casazza

Reviewer 1 ·

Basic reporting

The authors present a pretty comprehensive overview of the trihelix-family in soybean. The analysis includes standard analysis of the genomic localisation of trihelix genes, predicted protein domains and cis-regulatory elements in the promoter region of the 71 trihelix members. Furthermore, the expression of the trihelix factors after abiotic stress application is shown. The depicted figures and overviews are easy to understand and very illustrating. In general, the manuscript is worth publishing in Peerj after implementing a couple of improvements.

The scientific writing style is sometimes quite awkward (362, 403,…) and partly not in a professional and unambiguous language, in parts grammatically wrong. Therefore, please revise the manuscript on that point to achieve the standard level.

Here a few examples:

in the abstract!!!: line 22 replace present by presented, line 24 suggested -->suggests, line 26 provided-->provide, line 27 in responding--> in response to
Writing mistake in line 261 (similar mistake in line 202) “These results indicating” is wrong….better: “These results indicate” or “These results are indicating”. However, the word “indicate” is too strong in this context…. better would be “suggest”. line 269, replace “were displaying” by “displayed” (results should be presented in simple past), line 271 replace “in” by “after” or “following”. Line 272 delete “There were”. Line 279, replace “The results suggested……genes was…” by “The results suggest……genes is”. (Conclusions should be presented in simple present). Rephrase line 307 “That was because” is an awkward expression. Line 357 replace “This fact indicated” by “This finding suggests” (be careful with the use of the word “indicate”). Line 366, delete “As we all know” (it is colloquial speaking), line 401 replace “This evidence strongly indicated” by “This finding suggests”. Line 244, replace “were exhibited” by “exhibited” (it should be AKTIVE, not PASSIVE)

In the manuscript are plenty of additional mistakes of that kind.

in the conclusion paragraph 419-429: delete “In addition, the expression pattern of the trihelix 425 genes in different tissues and the cis-elements of the promoters were also analyzed.”--> that is not a conclusion….

Please revise carefully the manuscript and consider rephrasing.

Experimental design

The paper is very theoretical and based on the in silico analysis of previously published raw data. Consequently, the evaluation of the trihelix factor function is very hypothetically and based mainly on already published trihelix members and the presented qPCR analysis (which gives only a rough hint about a putative biological function/ participation).

1. Please merge Table S5-S8, Data presentation would be more clear in this way.

2. Figure S4: The significance test is missing (Student’s test)

Validity of the findings

The validity of the finding is ok.

1. In Figure S3, cis-regulatory elements responsive to Jasmonate, Ethylene and Salicylic acid are depicted, which suggest an expression of corresponding genes after biotic stress perception. In Arabidopsis, trihelix transcription factors have been shown being involved in the plant immunity and being part of the MAP kinase signalling pathway (ASR3, GTL1). I suggest to introduce and discuss this issue more in detail and to perform expression analysis of the 71 trihelix members or at least of a selection following flg22-application (and, or Jasmonate, ACC (precursor of Ethylene) and Salicylic acid treatment). This analysis would give the paper a more comprehensive overview of putative trihelix factor functions, besides the abiotic stress response.

2. In Arabidopsis, putative MAP kinase binding sites have been found in the trihelix members, thereby providing an important clue for the post-translational regulation by phosphorylation…. Can you identify MAP kinase binding sites in soybean trihelix factors? If yes…are they conserved and how is their distribution among the different groups?

3. Figure 8: Why these 12 trihelix genes have been chosen? any reason?

4. The indication in line 262 is incomplete. At least salicylic acid-responsive element and ethylene-responsive element refer to transcriptional control of the corresponding genes after biotic stress perception.

Reviewer 2 ·

Basic reporting

Comprehensive bioinformatic analysis of soybean trihelix transcriptional factors has been done in this paper. However, some of the details and analysis need to be made more accessible. The description of the results lacks the depth expected from a research article. They did phylogenetic analysis, protein motif and gene structure analysis, but there is very little biological function exploration. The legend of figures and supplemental figures was not well described. The English language should be improved for publication. Some examples include misspelling, no space between words, misuse of uppercase/lowercase.

Experimental design

Liu and colleagues used bioinformatic approaches to identify 71 trihelix transcriptional factors in soybean. They analyzed these gens and provide comprehensive genomic information including chromosomal distributions, gene duplication, motif compositions, gene structures and cis elements. They also did RNA-seq analysis of soybean plants with flood treatment and found a few of them are differentially expressed between conditions.

Validity of the findings

71 trihelix transcription factors have been identified in soybean. RNA-seq data from submergence experiment indicates few trihelix genes likely involved in flooding stress.

Additional comments

1, Line 64, analysis should be analyzed
2, Line 71-72, should make it clear that the data is from the publicly available RNA-seq database and in the result part the authors should cite the data source.
3, Line 133-134, what’s the q value?
4, Line 142 and 148, does root sample include all the root material or part or the root?
5, Line 165, what is the trihelix family size in other species
6, Line 164-190, Combine the two parts?
7, Trihelix factors also known as GT-factors because they bind to a typical GT domain. Could binding motif analysis be done?
8, Line 298-301, rephrase sentence
9, Line 306-309, rephrase sentence
10, please include reference for RNA-seq raw data in the text, figure legend and supplemental table.
11, Line 359-360, what are the two genes show different expression pattern? What the similar expression pattern that the 11 genes share?
12, Line 388-395, Literature was not well referenced. Xie et al (2009) reported that both GmGT-2A and GmGT-2B are involved in salt, freezing and drought stress, and GmGT-2B but not GmGT-2A affects ABA sensitivity.
For NaCl treatment, have the authors tried 150 mM?
13, Line 638-639, UTR is part of exon, not independent of exon.
14, Not all of the 15 genes are significantly differentially expressed genes.
15, What the difference between Figure 6 and Figure S6? It looks like that the same gene has different element.
16, Combine Supplemental table S4, S5, S6, S7 and S8

---

## Round 0.2 · Major Revisions

Dear Dot. Liu,

As you can see in the re-reviews one of the reviewers positively values your change. On the contrary, the other reviewer (R2) has significant concerns about your manuscript despite the changes. However, we would like to give you the opportunity to respond to their comments and criticisms. So, before re-submitting the manuscript carefully check that all weakness and errors are solved throughout the text.

Best regards
Gabriele Casazza

Reviewer 1 ·

Basic reporting

English spelling and style have been significantly improved.

There is a mistake in reference to Völz et al, 2018.(GTL1). In the reference list, the citation is included, but cannot be found in the main text body...please update that and check carefully the references again for additional mistakes.

Experimental design

ok

Validity of the findings

ok

Reviewer 2 ·

Basic reporting

The English language still need to be improved for publication. Article structure is good but some of figures are not shown in a professional format. Some of the references are not cited properly. There are errors and mistakes in this manuscript that are not acceptable for a scientific article.

Experimental design

The authors used bioinformatic approaches to identify 71 trihelix transcriptional factors in soybean and provided genomic information of these genes which could be useful. They also did RNA-seq analysis of soybean plants with flood treatment and found a few of them are differentially expressed between conditions.

Validity of the findings

This article provide some useful information of soybean trihelix factors.

Additional comments

Please read your manuscript and check all details carefully before you submit it.
Minor comments
1, Could you reference the soybean RNA-seq data itself but not Phytozome?
2, Could you please label the domain name in Figure 4? Figure 4 legend is not updated. “Figure S1” should be Figure S2 since you added another supplemental figure.
3, Xie et al (2009) did state that GmGT-2A showed a higher expression level at 12h after the ABA treatment, but it is a side effect. Also, the 2A overexpression lines did not show any significant difference compared to wildtype with ABA treatment. And Xie et al (2009) concluded that GmGT-2B affects ABA sensitivity.
4, Delete lines 357 to 362
5, Delete line 419
6, For Figure 9, could you use the same format for all the charts?

---

## Round 0.3 · Minor Revisions

Dear Dr. Liu,

The reviewer found the new version of the manuscript much improved. Nevertheless, they suggested some minor changes. So, I ask you to perform these few changes before the manuscript acceptance for publication.

Once again, thank you for submitting your manuscript to PeerJ and we look forward to receiving your revision. Please, respond point-to-point to the comments of reviewers to speed up the process of revision

Best regards
Gabriele Casazza

Reviewer 2 ·

Basic reporting

The manuscript has been improved. There are still some mistakes and errors.

Experimental design

No comment

Validity of the findings

No comment

Additional comments

1, Line 152 and 158, please rephrase it. All root material from 5 plants was pooled and three…
Technically, we don’t pool samples unless we can’t get enough material from one sample/plant. Because pooling can reduce the effects of biological variation. Instead, we include more than three biological replicates.

2, Line 185, cDNA should CDS. cDNA includes UTR.

3, Line 193-194, 215-217 etc., please use present tense. There are plenty of mistakes of this kind.

4, Line 211, spices should be species

5, line 238, delete much

6, line 293-294, what (9) and (13) are?

7, Please rephrase line 316

8, line 320-339, the genes that response to different stress treatment are still confusing. “11 genes” in line 318 are the same as that in line 320? Are all the “12 genes” in line 327 from the 15 DEGs? How did the authors select the 12 genes?

9, line 357, should be 3 h, 12 h.

10, line 364, Osorio et al (2012) found 63 trihelix genes in soybean. Do the 71 genes include all those 63 genes? What the other 8 genes are? Could you please provide information (e.g. gene organization, gene expression pattern etc.) of the genes that were not found by Osorio et al (2012)?

---

## Round 0.4 · accepted · Accept

Dear Dr. Zhang all changes solicited from the reviewer were done. So, I am very pleased to say that your paper " Genome-wide characterization and expression analysis of soybean trihelix gene family " is accepted for publication in the PeerJ. Congratulations!

Thank you for submitting your work to PeerJ.

Yours sincerely,
Gabriele Casazza